# Atrial Fibrillation Progression Is Associated with Cell Senescence Burden as Determined by p53 and p16 Expression

**DOI:** 10.3390/jcm9010036

**Published:** 2019-12-23

**Authors:** Laurence Jesel, Malak Abbas, Sin-Hee Park, Kensuke Matsushita, Michel Kindo, Hira Hasan, Cyril Auger, Chisato Sato, Patrick Ohlmann, Jean-Philippe Mazzucotelli, Florence Toti, Gilles Kauffenstein, Valérie Schini-Kerth, Olivier Morel

**Affiliations:** 1INSERM UMR 1260–Regenerative Nanomedecine, FMTS, Université de Strasbourg-Faculté de Pharmacie, 67401 Illkirch-Graffenstaden, France; Laurence.JESEL-MOREL@chru-strasbourg.fr (L.J.); Malak.abbas01@hotmail.com (M.A.); akqmf02@qmail.com (S.-H.P.); matsuken_22@yahoo.co.jp (K.M.); hira_heeray@hotmail.com (H.H.); cyril.auger@unistra.fr (C.A.); koge_bread1116@hotmail.com (C.S.); toti@unistra.fr (F.T.); gilles.kauffenstein@gmail.com (G.K.); 2Hôpitaux Universitaires de Strasbourg, Pôle d’Activité Médico-Chirurgicale Cardio-Vasculaire, 67000 Strasbourg, France; michel.kindo@chru-strasbourg.fr (M.K.); patrick.ohlmann@chru-strasbourg.fr (P.O.); jean-philippe.mazzucotelli@chru-strasbourg.fr (J.-P.M.)

**Keywords:** atrial fibrillation, senescence, aging, tissue factor, endothelial dysfunction, remodeling

## Abstract

Background: Whilst the link between aging and thrombogenicity in atrial fibrillation (AF) is well established, the cellular underlying mechanisms are unknown. In AF, the role of senescence in tissue remodeling and prothrombotic state remains unclear. Aims: We investigated the link between AF and senescence by comparing the expression of senescence markers (p53 and p16), with prothrombotic and inflammatory proteins in right atrial appendages from patients in AF and sinus rhythm (SR). Methods: The right atrial appendages of 147 patients undergoing open-heart surgery were harvested. Twenty-one non-valvular AF patients, including paroxysmal (PAF) or permanent AF (PmAF), were matched with 21 SR patients according to CHA2DS2-VASc score and treatment. Protein expression was assessed by tissue lysates Western blot analysis. Results: The expression of p53, p16, and tissue factor (TF) was significantly increased in AF compared to SR (0.91 ± 0.31 vs. 0.58 ± 0.31, *p* = 0.001; 0.76 ± 0.32 vs. 0.35 ± 0.18, *p* = 0.0001; 0.88 ± 0.32 vs. 0.68 ± 0.29, *p* = 0.045, respectively). Expression of endothelial NO synthase (eNOS) was lower in AF (0.25 ± 0.15 vs. 0.35 ± 0.12, *p* = 0.023). There was a stepwise increase of p53, p16, TF, matrix metalloproteinase-9, and an eNOS progressive decrease between SR, PAF, and PmAF. AF was the only predictive factor of p53 and p16 elevation in multivariate analysis. *Conclusions:* The study brought new evidence indicating that AF progression is strongly related to human atrial senescence burden and points at a link between senescence, thrombogenicity, endothelial dysfunction and atrial remodeling.

## 1. Introduction

Atrial fibrillation (AF) is associated with increased mortality due mainly to heart failure and embolic complications. AF is well known to occur more frequently with increasing age and is linked to vascular aging. During AF, inflammation, apoptosis, endothelial dysfunction, and platelet activation contribute to creating a prothrombotic state and promoting atrial remodeling [1]. While the link between aging and thrombogenicity is well established [2,3], the cellular and molecular mechanisms are still under consideration. Premature cellular senescence is an irreversible form of cell cycle arrest that can be triggered by various cellular stresses, including deoxyribonucleic acid damage, oxidative stress, and oncogene activation [4]. It is characterized by the acquisition of a proinflammatory and prothrombotic profile [5,6]. Senescent cells are found in aged tissues where they remain metabolically active but are unable to proliferate despite the presence of mitogens. Their role and effect in tissue is not well understood.

p53, a protein with a central role in DNA repair and cell cycle regulation, is pivotal in the induction of senescence [4]. Physiological p53 activity prevents cancer and protects from aging, whereas excessive p53 activation is detrimental. Increased p53 expression induced cardiac inflammation and altered systolic function in an animal model of heart failure [7]. Other data obtained in a murine model have highlighted a key role of p53 and p16 in the regulation of profibrotic signaling, such as premature senescence of myofibroblasts [8]. In AF, the role of senescence in atrial remodeling and the development of a prothrombotic state remains unclear. Using an original model of atrial endothelial cells, we recently demonstrated that thrombin, a key determinant of thrombogenicity during atrial fibrillation, promotes atrial endothelial cells senescence and the acquisition of the senescence-associated secretory phenotype, characterized by enhanced expression levels of vascular cell adhesion molecule (VCAM)-1, tissue factor, transforming growth factor (TGF-β), and metalloproteinases (MMP-2 and MMP-9).

In this study, we investigated the link between AF and senescence markers through the assessment of protein expression in the tissue lysates of human appendages from patients in AF and in sinus rhythm (SR). The protein expression levels of p53, p16 (a down-stream modulator of cell cycle arrest), tissue factor (TF, the cellular initiator of the coagulation cascade), MMP-9 (a matrix metalloproteinase involved in extracellular matrix remodeling), and eNOS (the pivotal endothelial NO synthase that participates in redox-induced senescence and endothelial dysfunction when down-regulated) were investigated. Immunohistological staining of p53, p16, and eNOS were also performed on appendage sections of AF and SR patients.

## 2. Material and Methods

### 2.1. Patients

Between October 2013 and October 2015, all patients undergoing open-heart surgery in our hospital (University hospital, Strasbourg, France) were screened for participation in this study. Surgery was mainly coronary artery bypass graft surgery or aortic valve replacement. Patients with mitral valve surgery were excluded to avoid valvular AF. One hundred and forty-seven patients gave informed consent in accordance with the local Ethics Committee from Strasbourg University Hospital (reference number EK2044 obtained the 4th of March 2013). Patients with relevant comorbidities (malignancies, chronic inflammatory diseases) were excluded.

Patients were classified according to heart rhythm and duration of AF in 3 groups: sinus rhythm (SR), paroxysmal AF (PAF), or permanent AF (PmAF). Out of the 147 patients included, 21 presenting with AF were matched with 21 SR patients, according to CHA2DS2-VASc score criteria (age, sex, hypertension, diabetes, vascular disease, congestive heart failure, stroke, or systemic embolism) and drug treatment (AT1 blockers, ACE inhibitors and statins). All patients underwent medical history recording, transthoracic echocardiography, Euroscore (European system for cardiac operative risk evaluation) calculation, 12-lead electrocardiography (ECG) recording, and routine hematologic and biochemical blood tests prior to the surgery.

Paroxysmal atrial fibrillation (PAF) was defined as self-terminating and recurrent AF of less than 7 days and permanent AF (PmAF) as chronic AF of at least 12 months duration with failed or unattempted cardioversion.

### 2.2. Sample Preparation

Right atrial appendages (RAA) were harvested during cannulation of the RAA for open heart surgery (a step performed to connect the patient to the heart-and-lung machine). Taking this heart sample carried no additional risk to the patient.

The samples were immediately minced and snap-frozen in liquid nitrogen for subsequent Western blot analysis. In addition, one piece was embedded in Tissue-Tek OCT (optimal cutting temperature) for subsequent immunofluorescence histological analysis. Frozen samples were stored at −80 °C.

### 2.3. Western Blot Analysis (WB)

Proteins were extracted from right atrial appendages, as described previously [9], and 30 µg total proteins were separated by SDS-PAGE (12% acrylamide, Euromedex, Souffelweyersheim, France) at 100 V for 2 h and transferred electrophoretically onto Polyvinylidene Difluoride membranes (PVDF, GE Healthcare Life Sciences, Brumath, France) at 100 V for 120 min. Aspecific binding sites were saturated by incubation of membranes with Tris-buffered saline solution (TBS, Euromedex) containing 3% bovine serum albumin and 0.1% Tween 20 for 1 h. Membranes were probed overnight at 4 °C with blocking solution containing a specific antibody directed against the protein of interest. The antibodies used were either mouse anti-TF (1:1000, Sekisui Virotech GmbH, Darmstadt, Germany), anti-eNOS (1:1000, BD Bioscience, San José, CA, USA), anti-human p16 (1:1000; Santa Cruz Biotechnology, Heidelberg, Germany), anti-human p53 (1:1000; Santa Cruz Biotechnology), or anti-human MMP-9 (1:10000 Abcam, Paris, France).

Membranes were incubated with an appropriate horseradish peroxidase-conjugated secondary antibody. The density signal of each band was detected using enhanced Clarity Western Electro Chemiluminescence Substrate (ECL, 170-0561, BIO-RAD, Marnes-la-Coquette, France). Membranes were incubated with a mouse polyclonal anti-beta tubulin antibody for normalization purposes. Results were expressed as beta tubulin density ratio using the ImageQuant acquisition system and analysis software (LAS4000, ImageQuant TL 8.1, Amersham, Les Ulis, France). The elevation of p53, p16, TF, eNOS, and MMP-9 were arbitrarily defined by values above the median (>0.7649 for p53, >0.4823 for p16, >0.7413 for TF, >0.3299 for eNOS, and >0.7430 for MMP-9)

### 2.4. Immunofluorescence Studies

RAA were embedded in Tissue-Tek OCT (#4583 Sakura Finetek SAS, Villeneuve d’Ascq, France) and were cryosectioned at 14 µm. Sections were air dried for 15 min and stored at −80 °C. Sections were fixed with paraformaldehyde at 4% (Electron microscopy sciences, Hatfield, PA, USA), washed, and treated with either 10% non-fat milk (Régilait, Saint-Martin-Belle-Roche, France) or 5% goat serum (Abcam) in Phosphate Buffer Saline (PBS) containing 0.1% Triton X-100 (Sigma Aldrich, Saint-Quentin-Fallavier, France) for 1 h at room temperature to block non-specific binding. RAA sections were then incubated overnight at 4 °C with an antibody directed against p53 (1:400, Santa Cruz Biotechnology), p16 (1:400, Santa Cruz Biotechnology), and eNOS (1/200, cat: 610297, BD Biosciences).

For negative controls, primary antibodies were omitted. Sections were then washed with PBS, incubated with the fluorescent secondary antibody (1/400, Alexa 633-conjugated goat anti-rabbit or anti mouse IgG (#A-21070 and #A-21050, Thermo Fisher, Illkirch-Graffenstaden, France) for 2 h at room temperature in the dark before being washed with PBS, mounted in Dako fluorescence mounting medium (#S3023 Dako, Agilent Technology, Les Ulis, France), and cover-slipped before being evaluated by confocal microscopy using a confocal laser-scanning microscope (Leica TSC SPE II, Mannheim, Germany). Quantification of fluorescence levels was performed using Image J software (version 1.49p, National Institutes of Health, Bethesda, MD, USA, https://imagej.nih.gov/ij/, 1997-2018).

### 2.5. Statistical Analysis

Continuous variables are expressed as mean ± standard error of the mean (SEM) or median (interquartile range) and categorical variables as frequencies and percentages. Continuous variables were analyzed for normal distribution using the Shapiro–Wilk test. Continuous variables between paired groups were compared using paired Student’s *t* test or by non-parametric test, as appropriate. Fisher’s exact test was used for comparison of categorical variables. To determine predictive factors of p53 or p16 elevation, the cohort was split into two subgroups according to p53 or p16 elevation (above or below the median value). Univariate and multivariate analysis of p53 or p16 elevation was done using bivariate logistic regression analysis. Variables with *p* < 0.05 in univariate analysis were entered into a stepwise ascending multivariate analysis. The results of the binary logistic regression are presented as hazard ratios (HR), their 95% confidence intervals (CIs), and *p* values. A *p* value < 0.05 was considered statistically significant. Statistical analysis was performed using SPSS version 13.0 software (SPSS Inc., Chicago, IL, USA).

## 3. Results

### 3.1. Patients Characteristics

Twenty-one patients with AF, including 11 PAF and 10 PmAF, were compared to 21 matched patients in SR. At the time of surgery, 4 PAF patients presented ongoing AF. Baseline characteristics of patients are summarized in Table 1.

Age, body mass index, sex, and cardiovascular risk factors were equally distributed in AF and SR groups, testifying that the 2 groups were homogeneous. Euroscore I and II were 1.5-fold more elevated in the AF group compared with the SR group without reaching statistical significance. The repartition of patients according to their CHA2DS2-VASc score was similar in the two groups.

There was no difference concerning drug treatment (statin, ACE inhibitors, AT1 blockers, nor β-blockers). As expected, patients from the AF group were more often under vitamin K antagonist treatment, whereas patients from the SR group were more often under aspirin treatment.

Echocardiographic parameters were similar between the 2 groups except the left atrium (LA) area, which was greater in the AF group. The biological parameters groups were similar in the 2 groups (Table 1).

### 3.2. p53, p16, TF, and MMP-9 are Upregulated and eNOS Down-Regulated in the Right Atrial Appendages of Patients with AF

WB analysis of the appendages lysates indicated that the expression of p53 and p16 were significantly increased in the AF group compared with the SR group (0.91 ± 0.31 vs. 0.58 ± 0.31, *p* = 0.001 and 0.76 ± 0.32 vs. 0.35 ± 0.18, *p* = 0.0001) as well as that of TF, the main cellular activator of the blood coagulation (AF: 0.88 ± 0.32 vs. SR: 0.68 ± 0.29, *p* = 0.045) (Figure 1). Similar levels of MMP-9 expression were detected. Conversely, a significant decreased expression of eNOS was observed in appendages from the AF group compared with those of SR group (Figure 1). The level of MMP-9 expression was correlated to age (*r* = 0.538, *p* < 0.001), left ventricular mass (*r* = 0.487, *p* = 0.002), and CHA2DS2-VASc score (*r* = 0.638; *p* < 0.001). The level of p53 was correlated to that of TF expression (*r* = 0.533, *p* < 0.001), but not to the CHA2DS2-VASc score nor to age. The level of p16 was correlated to that of p53 (*r* = 0.388, *p* = 0.011) and inversely to that of eNOS (*r* = −0.335, *p* = 0.034).

LA size was not correlated to either p53, p16, MMP-9, or TF, but was correlated to CHA2DS2-VASc (*r* = 0.359, *p* = 0.0048). No significant correlation could be evidenced between Euroscore 1 and 2 and p53, p16, and eNOS expression. Of note also, p53 and p16 markers were not different between genders.

Changes in protein levels, as assessed by WB, were confirmed by immunohistological analysis of the fresh-frozen OCT-embedded RAA sections. The level of p53 and p16 staining was pronounced in the AF group compared to the SR group. Conversely, the level of eNOS staining was decreased in the AF group compared to the SR group (Figure 2).

### 3.3. Changes in the Protein Expression Level of p53, p16, MMP-9, eNOS, and TF are Related to the Progression of AF

Protein expression in appendage lysates was evaluated by WB according to SR, PAF, and PmAF. The protein expression levels of p53, p16, MMP-9, and TF were significantly more elevated in PmAF compared to SR (Figure 3). A stepwise increase in p53, p16, and TF could be evidenced from SR to PAF and PmAF. Interestingly, MMP-9 was clearly elevated in PmAF compared to SR or PAF, data that was not evidenced by comparing SR and AF (Figure 1 and Figure 3). Conversely, the expression of eNOS significantly decreased gradually from SR to PmAF (Figure 3).

### 3.4. Predictive Factors of p53 and p16 Elevation

No relationship between age, cardiovascular risk factors, echocardiographic parameters, Euroscore, and either p53 or p16 expression could be established (Table 2 and Table 3). A history of AF (paroxysmal or permanent) and PmAF or AF at the time of surgery was evidenced as a predictive factor of p53 elevation by univariate analysis (Table 2). By contrast, paroxysmal AF at the time of surgery was not predictive of p53 elevation. As expected, p16, the down-stream cyclin-dependent kinase inhibitor of p53, was associated with enhanced p53 expression. By multivariate analysis, AF was the only predictive factor of the expression of either p53 or p16 (Table 2 and Table 3).

### 3.5. Predictive Factors of TF, eNOS, and MMP-9 Elevation

Additional analyses were performed to identify predictors of TF, eNOS, and MMP elevation. P53 was evidenced as the sole predictive factor of TF elevation (Appendix A). AF was associated with eNOS decrease (Appendix A). Age, hypertension, and CHADS2-VASc2 score were identified as predictors of MMP-9 elevation by univariate analysis. By multivariate analysis, hypertension was the sole predictive factor of MMP-9 elevation (Appendix A).

## 4. Discussion

The major findings of the study indicated that the progression of AF is strongly related to the human atrial senescence burden as determined by p53 and p16 expression. The stepwise increase of senescence (p53, p16), prothrombotic (TF), and proremodeling (MMP-9) markers observed in the right atrial appendages of patients in SR, PAF, and PmAF points toward multiple interactions in the human atrium that enhance the senescence burden, atrial extracellular matrix remodeling, thrombogenicity, and other putative mediators involved in the progression of AF.

### 4.1. Senescence Burden and Atrial Fibrillation

Although the link between aging and AF has been extensively characterized in epidemiological studies, little is known about the alteration of atrial tissue during senescence. While the mechanisms of cellular senescence has been explored ex vivo, it is only recently that the importance of this mechanism has been appreciated in cardiovascular diseases, such as in myocardial fibrosis or heart failure inflammation [5,7,8,10]. The putative role of senescence in AF has, up to now, attracted little attention. We therefore explored a new paradigm linking senescence, atrial remodeling, and substrate alteration that lead to AF onset and perpetuation. Kim et al., first reported an enhanced p21 expression together with the upregulation of proapoptotic genes in human appendages from patients in PmAF [11]. p21 protein, along with p16, is a cyclin-dependent kinase inhibitor that functions as a negative regulator of cell cycle progression [4]. p21 and p16 gene expression are tightly regulated by the tumor suppressor protein p53, which mediates the p53-dependent cell cycle G_1_ phase arrest in response to several stress stimuli [4]. In atrial endothelial cells, we recently demonstrated that thrombin and angiotensin II are potent agonists of senescence characterized by increased β-galactosidase activity together with p53, p21, and p16 overexpression [12].

Initially considered as a damage response phenomenon resulting in an irreversible cell cycle arrest, cellular senescence is also characterized by reduced migratory behavior and the appearance of distinct morphological and functional changes that could contribute to the impairment of cellular homeostasis. More recently, several studies have emphasized that senescent vascular cells also secrete proinflammatory and proremodeling factors, which may alter the tissue environment [13]. To underscore a possible link between senescence burden and the extent of AF, we first characterized p53 and p16 expression in human right atrial appendages. Increased levels of p53 and p16 atrial expression could be evidenced in patients with AF history, consistent with the enhanced p21 expression level previously described by Kim [11]. In our study, the stepwise increase in p53 and p16 expression between SR, PAF, and PmAF is suggestive of a link between senescence and AF progression. If endothelial atrial senescence is associated with phenotypical changes that could participate in atrial tissue remodeling through TGF-β, MMP-2, and MMP-9 overexpression and AF maintenance [12], we could not also exclude that tissue remodeling could also induce senescence.

To characterize predictive factors of senescent burden, as assessed by p53 and p16 expression, logistic regression analysis was performed. While no relationship between age, cardiovascular risk factors, echocardiographic findings, CHADS2-VASc2 score, and senescent markers could be established, a strong relationship between AF and senescence could be evidenced. In the setting of AF, atrial senescence can be induced by a number of factors, such as low or oscillatory shear stress promoting a downregulation of endothelial NO synthase and enhanced angiotensin II, hypoxia, inflammation, or redox-mediated stress. In line with this view, p53 was identified as the sole predictive factor of eNOS downregulation (Appendix A).

Up to now, controversies remain on the role of the p53/p16/p21 pathway in the regulation of cardiac homeostasis. “Clearance” of p16-positive cells in a mouse model was reported to attenuate age-related deterioration of the function of several organs, including the heart [14]. Xie et al., reported that premature senescence and the p16 pathway were associated with atrial fibrosis in AF and may have an antifibrotic role [10]. However, this latter study had enrolled patients with valvular AF patients with great left atrial diameter particularly in the persistent AF group. Other authors have pointed at the importance of the p53 pathway in the regulation of a profibrotic pattern, premature senescence of myofibroblasts being viewed as an essential antifibrotic mechanism [8]. Another study has suggested the implication of p53-induced inflammation in heart failure [7]. Whether this upregulation of the p53 signaling pathway is deleterious or confers cytoprotection of atrial tissue in AF remains to be investigated. Although the precise study of the mechanistic pathways linking senescence to atrial tissue remodeling in AF was far beyond the scope of the present study, our data clearly demonstrated a tight relationship between senescence burden and AF progression in non-valvular AF patients.

### 4.2. Senescence is Associated with Endothelial Dysfunction and Enhanced Tissue Factor Expression

Endothelial senescence is characterized by oxidative stress along with a pronounced downregulation of eNOS expression and endothelial formation of NO [2,4]. Complex interplay between senescence and oxidative stress are illustrated by the fact that reactive oxygen species (ROS) can directly induce telomere shortening, the initial step of cellular senescence or take part per se in the reduction of NO bioavailability [15]. In the present study, the inverse correlation between eNOS and p16 expression shows a possible relationship between senescence, oxidative stress, and endothelial dysfunction. The observation of a reduced eNOS expression as AF progresses is consistent with previous findings showing that AF causes atrial endocardial dysfunction, leading to decreased NO concentration and downregulation of eNOS [16], and that successful pulmonary vein isolation enables improvement of endothelial dysfunction as measured by flow-mediated vasodilation in AF patients [17]. In chronic AF, downregulation of eNOS and increased oxidative stress, mainly originating from upregulation of NADPH oxidase, have also been involved in electrical remodeling [18]. The reduced NO bioavailability induces a shortening of the action potential duration by modulating sodium, calcium, and potassium channels [18]. Oxidative stress also promotes a profibrotic state favoring fibroblasts differentiation into collagen-secreting myofibroblasts [19] that leads to LA remodeling. Besides its role in the control of vascular tone and endothelial survival, NO plays a determinant role in platelet aggregation and TF expression at endothelial surface, favoring procoagulant activity.

In the present work, the rise of atrial TF expression paralleled AF progression and indicated an enhanced prothrombotic state. Although the cause of TF elevation could not be established, the correlation between p53 and TF expression was another indication of the prothrombotic feature of endothelial senescence. In the setting of AF, recent data have emphasized the view that alteration in atrial hemodynamics caused by AF rhythm disturbances may promote prothrombotic activity through enhanced TF expression [20]. Together, these findings highlight the view of a progressive endothelial senescence and dysfunction that favors thrombotic propensity and AF progression.

### 4.3. MMP-9 Expression as a Surrogate Marker of Atrial Remodeling and AF

Many observational studies have pointed at an enhanced expression of MMP-2 and MMP-9 in atrial tissue or enhanced concentration in the peripheral blood of AF patients [21]. MMPs 2 and 9 are secreted by fibroblasts, endothelial cells, and cardiomyocytes [22] and belong to a class of proinflammatory and proangiogenic factors that control tissue remodeling by regulating extracellular matrix degradation and the release of growth factors, such as TGF-β1, that trigger a profibrotic signaling. The observation of an enhanced expression of MMP-9 in human atrial tissue from AF patients is consistent with previous findings by Gramley, showing an increased expression level of MMP-2 and MMP-9 in human right atrial appendages with AF progression [23]. Kato et al., also showed that elevated MMP-2 and longer AF duration increased the risk for difficulty in restoring SR in AF patients [24]. In an aging mice model, it has been observed that an enhanced MMP-9 concomitant with the development of diastolic dysfunction, as a surrogate marker of cardiac fibrosis, and MMP-9 deletion attenuated the age-related decline in diastolic function in part by reducing TGF-β1 profibrotic signaling [25]. In the present study, no correlation between senescent markers and MMP-9 expression could be established. MMP-9 levels were correlated to age, arterial hypertension, CHA2DS2-VASc score as a possible illustration of the complex network linking aging, hypertension, and atrial remodeling.

## 5. Study Limitations

Although we rigorously tried to match our study controls (AF vs. SR), we could not exclude that significant bias remained. However, it should be emphasized that baseline characteristics were not significantly different between groups. We also believe that the limited number of patients enrolled in the present study could have impeded the detection of significant relationships between factors in subgroup analysis (for example the absence of significant p16 elevation with permanent fibrillation, only in paroxysmal AF). Finally, owing to the observational nature of the present study, no definitive mechanistic insight on the link between AF, senescence, prothrombotic, and proremodeling patterns could be provided. The mechanisms of cellular senescence and the intracellular pathways involved remain largely unknown. As a consequence, the present data should mainly be considered as “hypothesis generating” and should be further investigated in appropriate models.

## 6. Conclusions

This study provided compelling evidence indicating that AF progression is strongly related to human atrial senescence burden as determined by p53 and p16 expression. The stepwise increase of senescence, prothrombotic, and proremodeling markers observed between SR, PAF, and PmAF points toward a possible network that links the human atrium, senescence burden, endothelial dysfunction, thrombogenicity, and atrial remodeling.

## Figures and Tables

**Figure 1 jcm-09-00036-f001:**
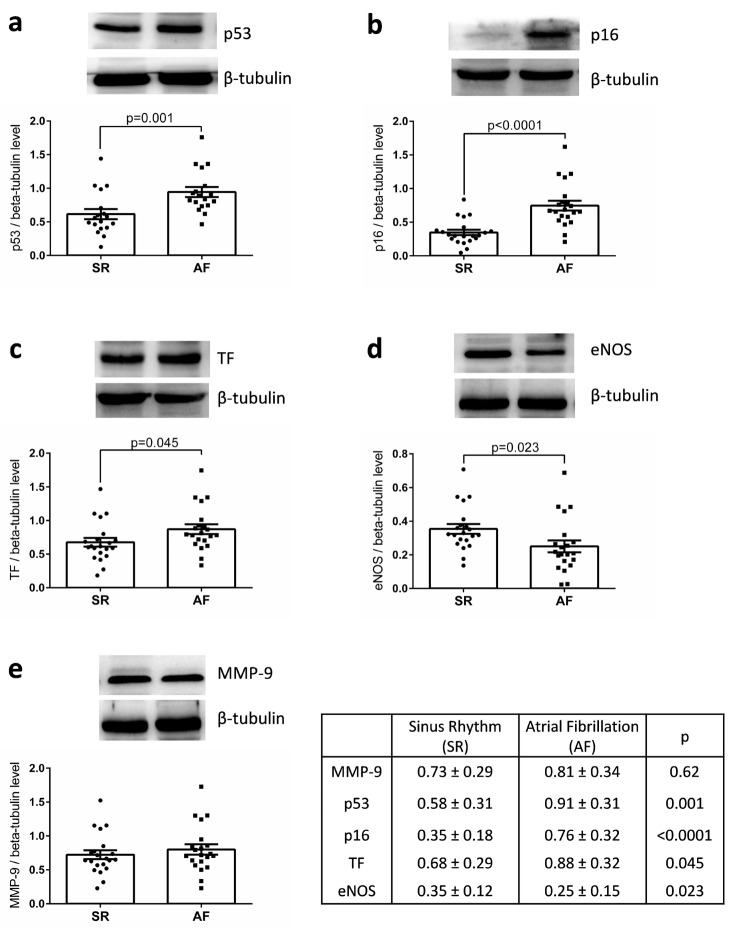
Expression of (**a**) p53, (**b**) p16, (**c**) TF, (**d**) eNOS, and (**e**) MMP-9 in right atrial appendages from patients in sinus rhythm (SR) or atrial fibrillation (AF). Protein expression was determined by Western blot analysis of 21 human right atrial appendages in each group. Results are expressed (mean ± standard error to the mean (SEM)).

**Figure 2 jcm-09-00036-f002:**
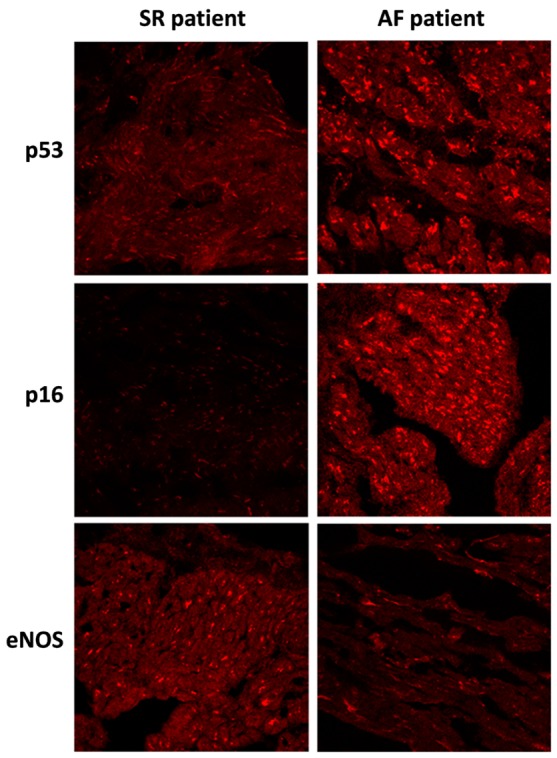
Immunofluorescence staining of p53, p16, and eNOS proteins in right atrial appendages sections from patients in sinus rhythm (SR) or atrial fibrillation (AF). Immunofluorescence staining of right atrial appendage sections were analyzed by confocal immunofluorocytometry.

**Figure 3 jcm-09-00036-f003:**
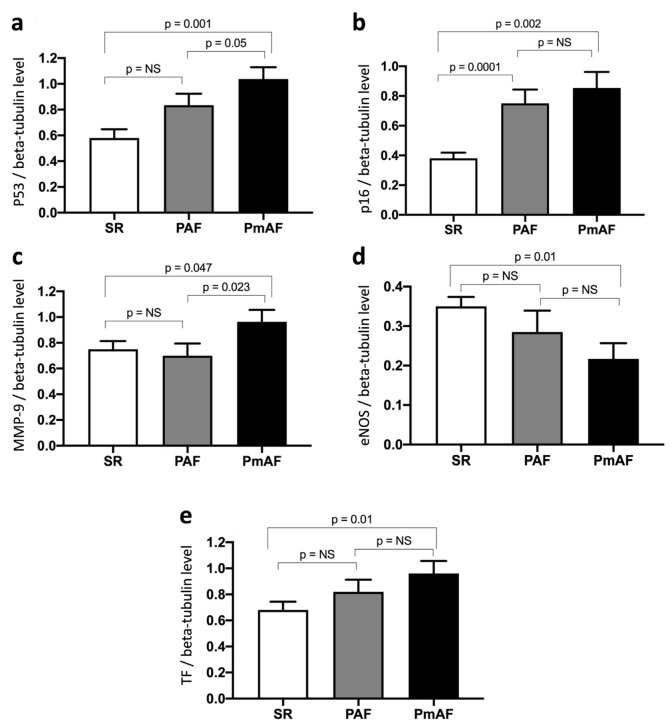
Expression of (**a**) p53, (**b**) p16, (**c**) MMP-9, (**d**) eNOS, and (**e**) TF in right atrial appendages from patients according to sinus rhythm (SR), paroxysmal (PAF), or permanent atrial fibrillation (PmAF). Protein expression was determined by Western blot analysis of 21 human right atrial appendages in SR, 11 in PAF, and 10 in PmAF. Results are expressed as mean ± SEM (NS, non significant).

**Table 1 jcm-09-00036-t001:** Baseline characteristics of patients from the sinus rhythm and the AF group.

Variable	Sinus Rhythm	Atrial Fibrillation	*p*
Age (years)	68 ± 11	70 ± 12	0.63
Sex (male)	13	13	1
Hypertension	14	13	0.75
Diabetes	7	8	0.75
Smoking	7	7	1
LVEF < 40%	1	1	
Vascular Disease	6	4	0.47
Body Mass Index	28.4 ± 5.7	30.5 ± 6.9	0.30
Euroscore I	4.2 ± 3.1	6.8 ± 5.1	0.06
Euroscore II	1.9 ± 1.8	3.5 ± 3.2	0.06
CHA2DS2-VASc			
0–1	6	5	
2–3	8	7	
≥4	7	9	
Drugs			
Beta-Blockers	12	15	0.33
ACE inhibitor/AT1 blocker	14	11	0.20
Aspirin	14	2	<0.001
VKA	2	14	<0.001
Statin	12	11	
Echographic Data			
LA Area (cm²)	23 ± 9	36 ± 19	0.02
RA Area (cm²)	18 ± 7	23 ± 9	0.15
LVEF (%)	62 ± 8	60 ± 8	0.41
EDLVD (mm)	51 ± 8	53 ± 10	0.50
LV Mass (g)	133 ± 41	127 ± 40	0.65
Biology			
Hb (g/L)	13.5 ± 1.1	13.5 ± 1.6	0.99
Leukocytes	6677 ± 2131	7191 ± 2004	0.43
Fibrinogen (g/L)	3.4 ± 0.5	3.6 ± 0.9	0.58
Creatinin Clearance (mL/min)	87 ± 30	78 ± 33	0.45

LVEF, left ventricle ejection fraction; ACE, angiotensin converting enzyme; AT1, angiotensin II type 1 receptor; VKA, vitamin K antagonist; LA, left atrium; RA, right atrium; EDLVD, end diastolic left ventricle diameter; LV, left ventricle.

**Table 2 jcm-09-00036-t002:** Predictors of p53 elevation: univariate and multivariate analysis.

Variable	Univariate Analysis			Multivariate Analysis		
	HR	95% CI	*p*	HR	95% CI	*p*
Age	1.011	0.959–1.066	0.68			
AF	10.240	2.475–42.370	0.001	7.849	1.330–46.333	0.023
Permanent AF	15.000	1.685–133.551	0.015			
Paroxysmal AF	1.600	0.413–6.193	0.49			
AF at the Time of Surgery	8.000	2.012–31.803	0.003			
Hypertension	1.231	0.38–4.358	0.75			
Diabetes Mellitus	0.533	0.148–1.922	0.34			
Smoking	0.538	0.422–5.606	0.51			
Female Sex	1.083	0.288–4.081	0.91			
LVEF <40%	2.105	0.176–25.170	0.56			
CHADS2-VASc	1.085	0.760–1.549	0.65			
Statins	0.699	0.433–1.129	0.14			
ACE inhibitor/AT1 blocker	0.667	0.190–2.334	0.53			
Coronary Artery Disease	0.300	0.083–1.081	0.07			
Euroscore I	1.089	0.937–1.266	0.27			
Euroscore II	1.063	0.844–1.338	0.60			
LA Area	1.032	0.973–1.094	0.30			
p16	16.121	1.461–177.851	0.023	1.986	0.109–36.312	0.643
eNOS	0.041	0.001-4.086	0.174			

AF, atrial fibrillation; LVEF, left ventricle ejection fraction; LA, left atrium, HR, Hazard Ratios.

**Table 3 jcm-09-00036-t003:** Predictors of p16 elevation: univariate and multivariate analysis.

Variable	Univariate Analysis			Multivariate Analysis		
	HR	95% CI	*p*	HR	95% CI	*p*
Age	1.010	0.959–1.065	0.70			
AF	18.062	0.871–84.283	<0.001	15.741	2.863–86.556	0.002
Permanent AF	3.000	0.655–13.747	0.16			
Paroxysmal AF	8.636	1.593–46.807	0.012			
AF at the Time of Surgery	13.600	3.091–59.831	0.001			
Hypertension	0.533	0.148–1.922	0.34			
Diabetes Mellitus	0.813	0.229–2.877	0.75			
Smoking	1.538	0.422–5.606	0.51			
Female Sex	0.686	0.182–2.589	0.58			
LVEF <40%	1.90	0.180–19.015	0.56			
CHADS2-VASc	1.050	0.737–1.498	0.79			
Statins	1.250	0.274–5.705	0.77			
ACE inhibitor/AT1blocker	0.284	0.076–1.063	0.06			
Coronary artery disease	0.677	0.198–2.312	0.53			
Euroscore I	1.036	0.899–1.194	0.62			
Euroscore II	1.091	0.861–1.383	0.47			
LA Area	1.028	0.972–1.087	0.33			
p53	7.842	1.167–52.711	0.034	1.532	0.133–17.610	0.73
eNOS	0.05	0.01–4.834	0.199			

AF, atrial fibrillation; LVEF, left ventricle ejection fraction; LV, left ventricle; LA, left atrium.

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
