# Peer review of "Atrial Fibrillation Progression Is Associated with Cell Senescence Burden as Determined by p53 and p16 Expression"

_jcm, 2019, doi:10.3390/jcm9010036_

Round 1
Reviewer 1 Report
The authors provide interesting data on the association between expression of markers of cellular senescence in atrial appendage and atrial fibrillation. The fact that most measured parameters show increased levels in patients with AF is intriguing and suggests and underlying pathogenic mechanism. The manuscript is well written and clearly presented.
a) Whilst the authors make clear that the data that is presented is only association and not causal in nature, this should be made a little more clear. In particular the title of the manuscript suggests that there is an impact of atrial fibrillation progression on senescence burden. This may be the case, but the data does not provide sufficient evidence for this. Association is all that can be concluded.
b) The absolute measurements of P53, p16 and other markers by Western blot analysis are clearly presented. In the statistical analysis, what cut-off has been used for p53 and p16 elevation in the univariate analysis. Multivariate data is not shown in tables 2 and 3.
c) On univariate analysis permanent AF is associated with p53 elevation. For p16 elevation, only paroxysmal AF is associated with this. How do the authors explain this?
d) Miscelaneous
Abstract AF “as compared to AF” and not “compared to” Full stop missing after reference 5 in 3rd paragraph of introduction. Tenses are mixed up in last paragraph of introduction. Would be better to say that protein expression “levels” of p53 etc “were” investigated. Same for last sentence of last paragraph of introduction. Immunohidtological stainings “were” performed. First paragraph of “Patients” section, don’t start sentence with a number (147), instead write One hundred and fourty seven. Table 1, missing abbreviations in legend; ACE, AT, VKA etc. Section 3.3 of results “data” and not “a data”. Conclusion compelling “evidence” and not “evidences” Font issues throughout manuscript, section 2.4 1st paragraph, section 4.1 1st paragraph, 4.3Author Response
Please see the attachment

Reviewer 2 Report
Review
Summary: The manuscript by L. Jesel et al. investigates the link between atrial fibrillation (AF) and senescence by comparing the expression of senescence markers (p53, p16) in atrial biopsies obtained from patients with AF and with sinus rhythm (SR). Authors report that expression of senescence biomarkers (p53, p16), tissue factor (TF), matrix metalloproteinase-9 increases during the progression of AF, while expression of endothelial NO synthase (eNOS) in AF is lower.
Broad comment:
Manuscript provides new data that level of several biomarkers that were linked to aging of the cells are increased in the patients with AF. One of the strengths of the paper, that data is obtained from the human samples.
However, the nature of the present study is highly observational and does not provide any insight on the links between atrial fibrillation and cellular senescence. Therefore, while this paper provides some ideas for future research, its impact is limited.
Specific comments:
Table 1. Do you have any insights why Euroscore I,II are so different between SR and AF patients? Could that affect interpretation of the results?Fig 1. Expression of proteins are normalized to beta-tubulin levels. Are the tubulin levels in SR and AF patients the same? It is well established that AF induces structural remodeling (increase in LA size, table 1) that involves changes in the organization of intracellular microtubulars and therefore might affect levels of tubulin expression.
Fig 1. No original blots are shown. In my opinion, examples of the original blots should be provided.
Fig 1. Title of “Y” axis reads: “xxx/beta-tubulin level (fold increase)”. I do not understand “fold increase” part of it. Increase over what? What is reference point?
Also, in Fig. 1 it would be beneficial if individual data points were provided, as that would give better representation on the data dispersion.
Fig 2. Immunostaining images do not match results shown in Fig.1 For example, from p16 staining one could get a wrong impression that SR patients have almost no p16 expression while it is skyrocketing in AF. The same goes to eNOS staining. How these examples were selected? Are these extreme cases or “typical” results? Could you provide some kind of florescence quantification?
If I understood correctly, in Fig.3 the same data set as in Fig.1 is shown. Just in this case AF group is divided into paroxysmal (PAF) or permanent atrial fibrillation (PmAF). I was just wondering, wouldn’t it make more sense to combine Fig.1 with Fig.3?
Table of predictive factors.
a) It should also include eNOS, TF and MMP-9;
b) Have you checked if there is correlation with male sex?
c) Why statistically significant results in Fig.1 and Fig.3 does not translate into the “predictive factors”? For example, PmAF does not predict p16 level (while this is highly significant change in Fig.3), or in the table PAF strongly predicts p53 increase, but the same change is insignificant in Fig. 3.
Paper title and discussion. Authors are trying to make a point that senescence contributes to AF development. However, this study does not provide any insights on the causal relationship and therefore it well might other way around, that AF remodeling is leading to the increase in senescence biomarkers. Such a possibility should not be excluded and has to be clearly stated.
Methods: description of western blots for eNOS is missing.
And finally, I think authors should use more caution when using of the term “senescence” in the context of this manuscript. Their results indeed show increase in p53 and p16, that also happen to be the biomarkers for aging cells. However, increase in senescence biomarkers does not automatically equal to “senescence”, which is much more complex process.
Round 2
Reviewer 2 Report
1) Fig 3.Title of “Y” axis still reads: “xxx/beta-tubulin level (fold increase)”. Please correct
2) Fig. 1 correct axis titles to “beta-tubulin”.
Also, on my screen Fig.1 has plenty of “?” symbols
3) Table of predictive factors:
To take into account the Reviewer’s criticism, study of eNOS as predictive factors of p53 or p16 elevation was added into tables.
I meant, that manuscript would benefit if you include separate tables with predictors for the expression levels for eNOS, TF and MMP-9 (in the same manner as you provided for p16 and p53).
